# Control of BKPyV-DNAemia by a Tailored Viro-Immunologic Approach Does Not Lead to BKPyV-Nephropathy Progression and Development of Donor-Specific Antibodies in Pediatric Kidney Transplantation

**DOI:** 10.3390/microorganisms13010048

**Published:** 2024-12-30

**Authors:** Michela Cioni, Stella Muscianisi, Marica De Cicco, Sabrina Basso, Hans H. Hirsch, Iris Fontana, Laura Catenacci, Jessica Bagnarino, Mariangela Siciliano, Oriana Montana Lampo, Gloria Acquafredda, Lou Tina Diana Boti, Jessica Rotella, Eleonora Bozza, Jennifer Zumelli, Kristiana Mebelli, Fausto Baldanti, Massimo Cardillo, Marco Zecca, Arcangelo Nocera, Mario Luppi, Enrico Verrina, Fabrizio Ginevri, Patrizia Comoli

**Affiliations:** 1Fondazione Malattie Renali del Bambino, IRCCS G. Gaslini Institute, 16147 Genova, Italy; michelacioni@gmail.com (M.C.); arcangelonocera@gmail.com (A.N.); enricoverrina@gaslini.org (E.V.); fabrizioginevri@gmail.com (F.G.); 2Transfusion Service, IRCCS G. Gaslini Institute, 16147 Genova, Italy; 3Cell Factory, Department of Mother and Child Health, Fondazione IRCCS Policlinico S. Matteo, 27100 Pavia, Italy; s.muscianisi@smatteo.pv.it (S.M.); m.decicco@smatteo.pv.it (M.D.C.); s.basso@smatteo.pv.it (S.B.); l.catenacci@smatteo.pv.it (L.C.); m.siciliano@smatteo.pv.it (M.S.); o.montanalampo@smatteo.pv.it (O.M.L.); g.acquafredda@smatteo.pv.it (G.A.); t.botilou@smatteo.pv.it (L.T.D.B.); j.rotella@smatteo.pv.it (J.R.); e.bozza@smatteo.pv.it (E.B.); jenniferzumelli95@gmail.com (J.Z.); k.mebelli@smatteo.pv.it (K.M.); 4Pediatric Hematology/Oncology, Fondazione IRCCS Policlinico S. Matteo, 27100 Pavia, Italy; m.zecca@smatteo.pv.it; 5Transplantation and Clinical Virology, Department of Biomedicine, University of Basel, 4009 Basel, Switzerland; hans.hirsch@unibas.ch; 6Vascular and Endovascular Department, Kidney Transplant Surgery Unit, Ospedale Policlinico San Martino, 16132 Genova, Italy; iris.fontana@hsanmartino.it; 7Microbiology and Virology, Fondazione IRCCS Policlinico S. Matteo, 27100 Pavia, Italy; j.bagnarino@smatteo.pv.it (J.B.); f.baldanti@smatteo.pv.it (F.B.); 8Transplantation Immunology, Fondazione Ca’ Granda, Ospedale Maggiore Policlinico, 20122 Milano, Italy; massimo.cardillo@policlinico.mi.it; 9Nephrology, Dialysis and Transplantation Unit, IRCCS G. Gaslini Institute, 16147 Genova, Italy; 10Section of Hematology, Department of Medical and Surgical Sciences, University of Modena and Reggio Emilia, AOU Modena, 41124 Modena, Italy; mluppi@unimore.it

**Keywords:** pediatric kidney transplantation, polyomavirus BK, cellular immunity, humoral immunity, donor-specific antibodies

## Abstract

Polyomavirus BK (BKPyV)-associated nephropathy (BKPyV-nephropathy) remains a significant cause of premature kidney allograft failure. In the absence of effective antiviral treatments, current therapeutic approaches rely on immunosuppression (IS) reduction, possibly at the risk of inducing alloimmunity. Therefore, we sought to explore the long-term effects of a tailored viro-immunologic surveillance and treatment program for BKPyV on the development of alloimmunity and kidney graft outcome. Forty-five pediatric kidney transplant recipients were longitudinally monitored for BKPyV replication, virus-specific immunity, and donor-specific HLA antibodies (DSAs). DNAemia developed in 15 patients who were treated with stepwise IS reduction. Among the other 30 patients, 17 developed DNAuria without DNAemia and 13 always resulted as BKPyV-negative. All patients with DNAemia cleared BKPyV after having mounted a virus-specific cellular immune response, and no biopsy-proven BKPyV-nephropathy was observed. The presence of cytotoxic populations directed to the BKPyV Large-T (LT) antigen early after transplantation protected kidney recipients from developing BKPyV replication, and the appearance of LT-specific T cells in viruric patients prevented the development of BKPyV-DNAemia. In our cohort, no significant correlation was observed between BKPyV-DNAemia and the development of DSA and antibody-mediated rejection. However, patients who experienced and cleared BKPyV-DNAemia had a worse allograft survival at a median follow-up of 18.9 years (*p* = 0.048). These data need to be confirmed in larger cohorts.

## 1. Introduction

Polyomavirus BK (BKPyV)-associated nephropathy (BKPyV-nephropathy) is one of the most challenging infectious complications of renal allograft dysfunction and graft loss [1,2,3]. To date, antiviral drugs with specific activity directed at the BKPyV life cycle are not available, and the therapeutic intervention of choice is immunosuppression (IS) reduction [2,3,4,5]. A significant improvement in BKPyV-nephropathy outcome has been observed following the adoption of systematic surveillance regimens and application of IS reduction at early stages of disease [6,7,8]. Following this approach, progression to proven BKPyV-nephropathy can be safely prevented if BKPyV-DNAemia is used to guide preemptive therapeutic intervention [9,10]. Since BKPyV-DNAemia has been shown to rapidly disappear by renal allograft nephrectomy in patients with continued immunosuppression such as kidney–pancreas recipients, its detection reflects allograft involvement [11]. Hence the terms probable and presumptive BKPyV-nephropathy reflecting increasing plasma BKPyV-DNA loads were proposed and proven valuable in clinical practice [12]. Among the multiple risk factors that contribute to BKPyV-nephropathy development, a central role has been assigned to a disruption in the balance between BKPyV replication and virus-specific immune surveillance [13,14,15,16,17]. Thus, in association with viral load determination in plasma [18], the quantification of BKPyV-specific immune responses could be used to characterize subgroups of patients at high risk of disease development. This combined risk classification would in turn improve individualized immune suppression decisions [19,20,21]. Indeed, BKPyV replication is generally an early event after allograft, and therapeutic IS reduction in this crucial phase of transplantation may induce acute rejection episodes or predispose to acute and/or chronic immune-mediated renal damage [22,23]. In this scenario, optimal modulation of immunosuppressive agents would produce the desirable reconstitution of BKPyV-specific immunity to control virus reactivation, while maintaining adequate immune suppression to protect the graft. In the present study, we analyzed the interplay between BKPyV replication, maintenance immunosuppression, and immune response to the virus, and their impact on kidney graft outcome, in a pediatric cohort of kidney recipients prospectively surveilled and treated with a stepwise IS reduction in the presence of presumptive or probable BKPyV-nephropathy [2,10]. The ultimate aim of this study was to assess the long-term effects of an individualized approach to BKPyV-nephropathy and related IS modulation on the basis of combined viremia and virus-specific immunity.

## 2. Materials and Methods

### 2.1. Patients

In this single-center study, patients referred to the Pediatric Nephrology Unit of the G. Gaslini Institute-Genova for kidney transplantation (KTx) and post-transplant care between December 2002 and June 2007 were analyzed. The demographic information and the baseline clinical parameters of the study subjects are provided in Table 1. Early BKPyV clinical and immunological data from this cohort have been previously published [10], while this study provides long-term follow-up data on BKPyV and DSA surveillance.

Baseline IS protocols adopted for the patients enrolled into the study included induction with the anti-CD25 monoclonal antibody basiliximab, and double or triple therapy with either cyclosporine-A (CyA) or tacrolimus (FK) together with prednisone alone or prednisone and mycophenolate mofetil (MMF). Three sensitized KTx patients received anti-tymocyte globulin (ATG) induction. Standard of care included a renal biopsy for graft dysfunction, which was defined as serum creatinine level elevation ≥ 20% than baseline. Treatment of acute rejection consisted of pulse steroids and, in case of steroid resistance (*n* = 1), photopheresis and switch to tacrolimus were employed as rescue therapy. All patients received antiviral prophylaxis with acyclovir for the first 6 months post-transplant, and pneumocystis prophylaxis with cotrimoxazole for 12 months.

Graft biopsies were performed for clinical indication (BKPyV-DNAemia, graft function decline, and/or proteinuria); since 2010, DSA positivity was also included among indications. Biopsies were histologically graded following Banff 97 criteria with Banff 2013 and 2017 updates [24]. Patients developing antibody-mediated rejection (ABMR) were treated with a protocol including a combination of plasmapheresis, i.v. human Ig, and anti-CD20 monoclonal antibody.

### 2.2. HLA Typing and Detection of Anti-HLA Antibodies

HLA class I and class II typing were performed as previously described [23].

Anti-HLA class I and class II IgG antibodies were tested by means of a bead-based detection assay, with LABScreen Mixed kit and Single Antigen kit (One Lambda, Canoga Park, CA, USA). The MFI cut-off for positivity was set at 1000 [25]. All sera were EDTA-treated before testing.

### 2.3. BKPyV Viral Monitoring

Blood and urine samples were collected during routine laboratory testing at 1, 3, 6, 9, 12, 18, 24, 36, 48 months post-transplantation.

#### 2.3.1. Serology Methods

Patients’ serostatus was determined by testing 1:400 diluted plasma in an IgG ELISA format with 50 ng of BK virus-like particles (VLPs) as antigens coated onto solid phase purified from major viral protein (VP1)-expressing baculovirus-infected SF9 cells after lysis and gradient centrifugation. A cut-off of optical density (OD) 492 nm > 0.110 after subtracting non-VLP-expressing lysates was considered positive [26]. Baseline BKPyV serology at transplantation was evaluated for all patients.

#### 2.3.2. Polymerase Chain Reaction (PCR) Methods

For the purposes of this study, qualitative PCR for the detection of BK DNA was used for patient management. BKPyV DNA detection was performed by a previously described nested, qualitative PCR assay [18,27]. Samples found positive for BKPyV DNA with the nested PCR were quantitated by a BKPyV-specific real-time PCR, according to a previously reported method [10,18]. BKPyV-DNAuria was defined as the presence of BKPyV DNA in urine, while BKPyV-DNAemia was defined by the detection of viral DNA in serum. Patients with BKPyV-DNAemia underwent a renal biopsy to evaluate the presence of BKPyV, and, in the absence of histological findings identifying definitive BKPyV-nephropathy, were diagnosed as probable/presumptive BKPyV-nephropathy [2,3]. Patients with probable BKPyV-nephropathy were initially monitored more frequently and treated with immunosuppression (IS) reduction only if viral load increased to ≥10^4^ copies/mL, while patients with presumptive BKPyV-nephropathy were preemptively treated with IS reduction, according to a protocol previously described [10]. Stepwise reduction in immunosuppression started with a 15–20% decrease in calcineurin inhibitor (CNI) plasma levels. In the case of increasing viral load over the next four weeks, MMF was halved as a further step or discontinued as a final step. Reduction in IS therapy was the only therapeutic intervention used to treat BKPyV-DNAemia in this cohort.

### 2.4. BKPyV-Specific Immune Monitoring

A total of 414 blood samples were collected and stored from the patients at the same time points for virological evaluation, with a sampling range of 5–15, according to the length of follow-up. All samples were processed within 24 h from the blood draw, and cryopreserved. Immunological monitoring of a single patient was performed by thawing and plating all samples on the same day in a single assay, in order to minimize interassay variability. BKPyV-specific T cell immunity was evaluated by measuring the frequency of virus-specific IFNγ-secreting cells in an ELISPOT assay, and cytotoxicity was measured in BKPyV-specific cell cultures, using a 51-chromium release assay. Peripheral blood mononuclear cells (PBMCs) were collected at different time points and cultured for 8–10 days in the presence of 15 mer peptide pools spanning the entire BKPyV-VP1 and large T (LT) proteins (0.5 μg/mL concentration for each single peptide, JPT Peptide Technologies, Berlin, Germany). ELISPOT assays were performed according to a previously described method [13]. Cultured T cells were seeded in the absence or presence of VP1 and LT peptide mixes (0.5 μg/mL). Response to phytohemagglutinin (PHA, 4 μg/mL) was employed as a positive control. After incubation for 24 h at 37 °C, plates were processed according to a standard procedure. IFN-γ-producing spots were counted using an Elispot reader (Bioline, Torino, Italy). The number of spots per well was calculated after subtracting assay background, quantitated as an average of 24 wells containing only sterile complete medium, and specific background was quantitated as the sum of cytokine spots associated with responders alone. Specific cytotoxic activity was assessed by a standard ^51^Cr-release assay, against a panel of targets including autologous PHA blasts pulsed for 2 h with 2 μg/mL of VP-1 and LT peptide mix or with 2 μg/mL of control peptide (EBV-LMP2 peptide mix, JPT), and incubated overnight with ^51^Cr (100 μCi). In brief, PBMCs cultured for 8–10 days with VP-1 and LT peptide pools were incubated with 1000 target cells at E:T ratios of 20:1, 10:1, 5:1, 2.5:1.

### 2.5. Statistical Analysis

Data were expressed as the median and range, or median and 95% confidence interval (CI) as appropriate. Comparison of immunological parameters among different subgroups was performed with the Kruskal–Wallis one-way analysis of variance. The correlation of immunological parameters with BKV viruria and viremia was evaluated with the Wilcoxon test. Event-free survival was estimated with the Kaplan–Meier method and comparison between risk groups performed with the log-rank test. For graft failure, the censoring event was death with a functioning graft, and graft failure due to relapse of end-stage renal disease. Patients who did not experience graft failure were censored at the end of follow-up. To determine differences among groups, we compared categorical variables with the Fisher’s exact test. *p* values < 0.05 were considered statistically significant; *p* values from 0.05 to 0.1 were not considered significant, but reported in detail, while *p* values > 0.1 were reported as non-significant (ns). Statistical analyses were performed using the NCSS System (NCSS, Cary, NC, USA).

## 3. Results

### 3.1. Study Population

A total of 45 patients, enrolled in a prospective BKPyV surveillance program [10] for whom suitable post-transplant cellular samples were obtained, and with longitudinal serum samples for DSA analysis, were evaluated for BKPyV-specific cellular and humoral immunity, and for DSA development (Figure 1).

Of the 45 patients, 17 experienced urinary shedding in the absence of viremia (group 2), 15 developed concomitant viremia (group 3), while 13 were always BKPyV-negative (group 1) at the perspective surveillance time points (Figure 1). Among the patients with urinary shedding alone, nine had sustained BKPyV-DNAuria (at least two consecutive samples positive for BKPyV DNA; median peak DNAuria: 2.7 × 10^8^), while eight had transient DNAuria (single BKPyV DNA positivity; median peak DNAuria: 1.2 × 10^5^). Median BKPyV-DNAemia onset was at 3 months post-transplant, and median duration was 2 months (range 1–9 months). Nine of the fifteen patients had peak viremia > 10^4^/mL, which lasted a median of 3 months. None of the patients included in the study were diagnosed with proven BKPyV-nephropathy, with a median observation period of 18.9 years (3.5–21.7 years). None of the patients had BKPyV-DNAemia in the absence of BKPyV-DNAuria. A total of 9 out of the 45 patients included in the study were BKPyV-seronegative at transplantation (20% of the total population, distributed as follows: 15% in the BKPyV DNA-negative group, 18% in the positive DNAuria group, and 26% in the BKPyV-DNAemic patients, *p* = ns).

### 3.2. Kinetics of Humoral and Cellular Immunity to BKPyV After Kidney Transplantation

In order to prospectively assess cellular and humoral immunity to BKPyV, the frequency of IFN-γ-producing cells, specific cytotoxic capacity, and specific IgG levels were evaluated in samples obtained from the patients at different time points after transplantation, and the results were then analyzed in relation to the BKPyV status of the patient at the time of evaluation (Figure 2). After transplantation, the frequency of BKPyV-specific IFNγ-producing cells increased according to the degree of viral exposure. In detail, Vp1-specific T cells increased in the three groups (Figure 2A). Conversely, LT-specific T cells, which were high (median 70 spot-forming units SFU/10^5^ cells) and remained unchanged throughout the follow-up period in patients belonging to group 1, had a significant increase in recipients belonging to both groups 2 and 3 (Figure 2B). Similar kinetics were observed for BKPyV cytotoxicity. In groups 2 and 3, cytotoxic T cell responses to Vp1 and LT antigens were mostly absent before exposure, and an efficient lysis against BKPyV antigen-bearing targets was observed only in viremic patients following therapeutic IS reduction (Figure 2C,D). However, patients belonging to group 1 had significantly higher cytotoxicity to LT antigens at 1 month after transplantation, compared to patients belonging to groups 2 and 3 (median cytotoxicity in group 1: 8% vs. 0% in groups 2 and 3, *p* < 0.05). Among the three BKPyV replication groups, we also evaluated the immunological behavior of BKPyV-seronegative patients versus BKPyV-seropositive at transplantation. We did not observe significant differences in the response kinetics to BKPyV in the latter cohorts. In particular, BKPyV-seronegative patients who developed BK viremia after transplantation had comparable levels of cellular immunity at peak viruria, peak viremia, and at viremia clearance in comparison to BKPyV-seropositive viremic patients. In the case of humoral immunity, BKPyV-seropositive patients who never reactivated the virus did not show a significant increase in IgG levels (from a median OD of 0.37 at month + 1 to 0.39 at the end of follow-up), while patients with urinary shedding alone or with viremia increased from a median OD of 0.3 to 1.2 (*p* = 0.09), and 0.26 to 2.8 (*p* < 0.0005) (Figure 2E). At peak DNAuria, BKPyV-DNAemic patients already exhibited an 8-fold rise in specific IgG compared to the 1.5 increase observed in group 2 recipients.

### 3.3. Immunological Predictors of BKPyV Replication and Reactivation

We first determined whether a general state of immune deficiency, likely due to an excess of immune suppression, could account for the development of BKPyV reactivation. Thus, we assessed whether there was a correlation between T cell immunity, measured as the production of IFNγ in response to stimulation with a polyclonal T cell activator such as PHA, and the presence of BKPyV DNAuria and DNAemia. Development of BKPyV replication, measured as BKPyV urinary shedding, did not correlate with an impaired cellular response to PHA, as patients belonging to groups 2 and 3 showed measurable levels of PHA-directed IFNγ-SFU prior to BKPyV-DNAuria appearance, comparable to those observed in group 1 at +1 month after transplantation (median SFU/10^5^ cells: 255 in group 1 vs. 268 in group 2 vs. 232 in group 3) (Figure 3A). Moreover, the levels of cellular response to PHA did not correlate with cellular immunity to BKPyV VP1 and LT (Figure 3B,C).

We then proceeded to analyze the role of the different virus-specific cellular and humoral response compartments, namely cytokine production and cytotoxicity against Vp1 and LT antigens, and specific antibody production, in the development and progression of BKPyV replication. We observed that the only parameter that correlated with protection from the development of BKPyV replication was the presence of LT-specific cytotoxic activity early after transplantation. In detail, patients belonging to group 1 have significantly higher levels of LT-specific cytotoxicity at month + 1 from transplant, in comparison to allograft recipients included in both groups 2 and 3 (median cytotoxicity group 1: 7% vs. 0% in groups 2–3, *p* < 0.05) (Figure 4). Regarding the risk of progression from urinary shedding to BKPyV reactivation, patients with urinary shedding who do not progress to DNAemia show a median 3-fold increase in LT-specific T cell levels at peak DNAuria, compared to no increase observed in patients who develop DNAemia (Figure 5A). The latter group mounted a significant response to LT only after therapeutic IS reduction. As for BKPyV replication development, the response to PHA was not found to be significantly correlated with progression from BKPyV-DNAuria to DNAemia (Figure 5B). Following IS reduction, the appearance or a significant increase in BKPyV-specific immunity was observed in all patients, with the clearance of BKPyV-DNAemia after a median of 2 (range 1–9) months.

Upon the appearance of/increase in BKPyV immunity, stepwise IS reduction was stopped. The observation of a trend in viremia decrease prompted the resumption of the maintenance IS target levels, in order to limit prolonged suboptimal immunosuppression. Indeed, no difference in the CNI levels in the DNAemia-positive versus DNAemia-negative patients was observed at any time point during the post-transplant follow-up (Table A1). Likewise, no statistical difference was observed in memory immune responses to BKPyV or to PHA at 12 months or at the time of steady-state IS among patients with or without BKPyV-DNAemia, despite patients with BKPyV-DNAemia having higher frequencies of memory T cells (Table A2).

### 3.4. Immunological and Clinical Parameters of the Cohort and Their Correlation with Graft Outcome

Twenty patients (44%) developed DSA at a median time of 55 months (range 6–116 months) (Figure 1). Among the DSA-positive patients, six, seven, and seven developed anti-class I (anti-A in four patients, anti-B in four patients, and anti-C in one patient; median MFI peak of dominant antibody: 4800), anti-class II (anti-DQ in all patients; median MFI peak of dominant Ab: 17,020), or both anti-class I and II DSA (anti-A in six patients, anti-B in three patients, anti-C in one patient, anti-DR in two patients, anti-DQ in seven patients; median MFI peak of dominant Ab: 19,720), respectively. All patients with anti-class II DSA had at least one positivity for DQ antibodies. Most DSA+ patients showed prolonged positivity, which persisted throughout the follow-up, but one patient exhibited a transient DSA. Antibody-mediated rejection (ABMR) was diagnosed in 16 patients at a median follow-up of seven years. We analyzed the association between probable/presumptive BKPyV-nephropathy and onset of DSAs and ABMR in this cohort. No correlation was found between the presence of BKPyV-DNAemia and DSA (DSA+ 6/15 DNAemic vs. 14/30 non DNAemic pts; *p* = 0.75) or ABMR (ABMR+ 8/15 DNAemic vs. 8/30 non DNAemic pts; *p* = 0.1). Fifteen patients lost their graft at a median time of 6.7 years (range 3.5–14.5), due to ABMR (n = 11), chronic T cell-mediated rejection (TCMR) (n = 1), focal segmental glomerulosclerosis recurrence (n = 2), or amyloidosis (n = 1). None of these patients’ biopsies showed signs of BKPyV-nephropathy. We conducted a univariate analysis to assess the clinical and immunological parameters correlated with graft loss (start of renal replacement therapy) in our population (Table 2).

We demonstrated that the presence of DSAs, ABMR, and TCMR were significantly associated with graft loss. Regarding BKPyV replication, patients that developed DNAemia, but not those that remained DNAuric without progression to probable or presumptive BKPyV-nephropathy, experienced a significantly lower graft survival at long-term follow-up (Figure 1). Immunity to BKPyV or PHA at 12 months for BKPyV-negative patients or at maintenance IS steady state after BKPyV positivity did not correlate with graft loss (Table 2). Moreover, among viremic patients, the extent of cellular immunity to BKPyV or PHA at follow-up as described above did not correlate with graft loss [median response (and 95% CI) to BKPyV in patients without graft loss 341 SFU/10^5^ (118–543) vs. 446 (63–875) in patients experiencing graft loss, *p* = ns; PHA in patients without graft loss 444 SFU/10^5^ (102–540) vs. 533 (96–705) in patients experiencing graft loss, *p* = ns].

## 4. Discussion and Conclusions

Data on the long-term impact of BKPyV-DNAemia on kidney graft outcomes are limited. In this study, we provide a detailed clinical, virological, and immunological report of a pediatric kidney cohort followed prospectively for BKPyV events, who were treated with stepwise IS reduction for probable and presumptive BKPyV-nephropathy. With a median follow-up of 18 years, we observed that (i) close viral and immune monitoring and IS reduction were able to clear BKPyV-DNAemia and prevent BKPyV-nephropathy progression; (ii) BKPyV-DNAemia and our treatment protocol did not increase the risk of DSA or ABMR in our patient population; and (iii) despite successes in preventing humoral alloimmunity in our BK viremic patients, BKPyV-DNAemia negatively influenced long-term graft outcomes.

Our observation that BKPyV-DNAemia was not associated with DSA development is in line with a study conducted in a large cohort of adult kidney recipients, in which BKPyV-nephropathy, but not BKPyV-DNAemia, was an independent risk factor for DSA and subsequent ABMR [28]. In contrast, two other studies found a relationship between BKPyV-DNAemia and DSAs [29,30]. The discrepancies are likely due to different IS reduction protocols for BKPyV-DNAemia. Indeed, our treatment protocol for BKPyV-DNAemia was based on the reduction in CNI levels before the reduction in MMF, and only four patients from our cohort did reduce MMF, similarly to the viremic patients described by Cheungpasitporn et al. [28], while in the other studies, a large proportion of patients reduced or discontinued MMF [29,30]. Moreover, longitudinal BKPyV cellular immunity monitoring allowed for cautious IS reduction and relatively rapid resumption of maintenance IS in our patients, successfully preventing progression to BKPyV-nephropathy and related graft loss while avoiding the development of humoral alloimmunity.

In the study by Sawinski et al., the increased incidence of DSAs after BKPyV-DNAemia was not associated with a worse graft outcome at a median follow-up of 3 years [30]. In our population, as expected in the absence of an increased risk of DSAs, graft loss at 5- and 10-year follow-up was not significantly different in BKPyV viremic vs. non-viremic patients, as described in a cohort treated with preemptive IS reduction for BKPyV-DNAemia [22]. However, we were able to expand our observation further because of the long follow-up in our cohort, and we were surprised to find that at almost 19-year follow-up, kidney grafts from BKPyV viremic patients had a significantly worse outcome compared with kidney recipients that did not develop BKPyV-DNAemia. When looking at the outcome, the main cause of graft loss was ABMR, but BKPyV-DNAemia in our patients was not in itself associated with DSAs or ABMR. Moreover, when looking at the IS levels during the first year post-transplant and at steady state, BKPyV viremic patients did not differ from BKPyV-DNAemia-negative kidney recipients. We hypothesize that our finding may be related to subtle inflammatory/fibrotic damage to the graft caused by BKPyV, as described by Drachenberg et al. [31]. The lesions may not be so evident, as the early diagnosis allowed prompt viremia control and none of the patients were found with a histologic diagnosis of BKPyV-nephropathy, but they may have rendered the graft more susceptible to subsequent antibody damage in those patients that ultimately developed DSA and ABMR. Indeed, it is possible that the initial damage may have exposed cryptic antigens within the graft, favoring not only HLA-DSA formation but also non-HLA DSAs [32]. The statistical data are weak, likely due to the limited number of patients analyzed, and for this reason, the finding of a worse long-term graft outcome in KTx recipients with previous BKPyV DNAemia may be biased. Therefore, our results need to be confirmed in larger cohorts.

Our data suggest that even when optimizing a management protocol for BKPyV replication, and tailoring IS modulation, the long-term outcome of the grafts may be affected by probable/presumptive BKPyV-nephropathy. Thus, a different approach may be needed to manage BKPyV replication and prolong graft outcome, especially in the pediatric population with its long survival expectancy. Data from our and other groups indicated that the frequency of BKPyV-specific T cells and levels of specific IgG increased according to the degree of viral exposure, and that IS tapering in BKPyV-DNAemic kidney recipients led to an increase in BKV-specific T cell responses and viremia clearance [10,13,16,20,33,34,35,36,37]. The presence of defects that impair innate immunity may allow early viral spreading in the renal tubule of the allograft [38]. In line with others, we observed that the frequency of BKPyV-specific T cells, although generally low early after allografting, is higher in BKPyV-seropositive KTx recipients who do not develop detectable levels of BKPyV-DNAemia and/or BKBKPyV-nephropathy after transplantation [39]. The protective effect of virus-specific T cells is also suggested by the observation that viruric patients who do not progress to BKPyV-DNAemia are better able to respond to viral shedding by mounting a vigorous virus-specific cellular response. Moreover, we expanded prior evidence demonstrating that the loss of BKV-specific T cells directed to LT antigens from pre- to post-KTx-characterized patients are at increased risk of BKPyV replication [39], by showing that the presence of cytotoxic populations directed to LT early after transplantation protects kidney recipients from BKPyV-related disease. These populations are likely CD8+ T cells, as cytotoxicity was visible at 5 h, a temporal characteristic compatible with CD8+ effectors. The crucial role played by cytotoxic CD8+ T cells in preventing BKPyV-nephropathy progression has been previously underlined by studies demonstrating that CD8+ T lymphocyte functional impairment is correlated with clinically relevant BKPyV replication [20,36].

One way to boost BKPyV-specific immunity in patients positive for BKPyV-DNAemia is to act on CNI maintenance levels, as we did in our patients and which has been suggested by other studies [10,40,41]. However, in the early post-transplant period, balancing IS modulation is a difficult task that may lead to humoral alloimmunity. Our own and previously published data on the effects of early post-transplant BKPyV-specific immunity on the prevention of BKPyV-DNAemia suggest that an alternative approach should include the prevention of BKPyV viruria/viremia by administering BKPyV-targeted cytotoxic T cells [42,43,44].

Short LT-immunodominant peptides have been identified and employed to evaluate the effects of BKPyV-specific CD8+ T cells in KTx recipients with BKPyV replication [35,45]. The same peptides could be employed to expand cytotoxic T cell populations that could be used in a cell therapy strategy of BKPyV-nephropathy prevention, as has been performed for JCPyV-associated progressive multifocal encephalopathy [46,47]. It may be argued that T cells administered during the early phases post-transplant, when maintenance IS is at its highest, could fail to expand and control viral outgrowth. However, we believe that cultured cytotoxic effectors would still be able to exert their activity, although maybe with low expansion potential, as demonstrated with T cell therapy for post-transplant Epstein–Barr virus-related lymphoproliferative disease after KTx [48]. In addition, preclinical studies have demonstrated the feasibility of producing pathogen-specific T cells resistant to steroids [49], or to CI [50,51], and clinical trials are underway. Finally, as T cells from KTx recipients may have functional impairment [20,36], a viable option could be the use of third-party allogeneic T cells [48,52,53,54], perhaps with improved clinical and virological efficacy [55] and in combination with administering high neutralizing antibody activities [56].

In conclusion, despite a tailored strategy that allowed for BKPyV-DNAemia clearance and the prevention of BKPyV-nephropathy progression while avoiding the development of humoral alloimmunity, KTx recipients who developed BKPyV-DNAemia may experience worse long-term graft outcomes. Evaluation of BKPyV-specific immunity early after transplantation, and the promotion of virus-specific cellular therapy in allograft recipients with high-level BKPyV-DNAuria, may prevent the onset of BKPyV-DNAemia and ameliorate outcomes.

## Figures and Tables

**Figure 1 microorganisms-13-00048-f001:**
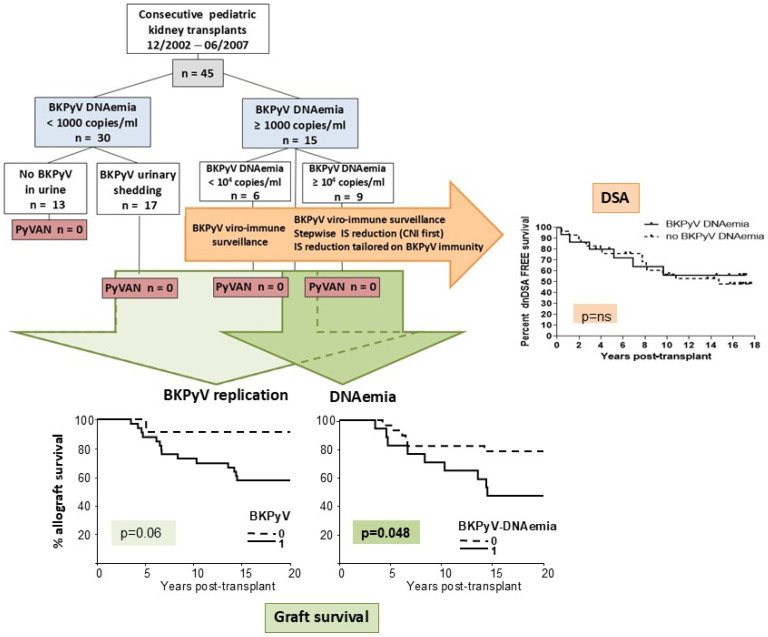
Flow diagram of study cohort and main outcome results for BKPyV infection, DSA, and graft survival. Left-bottom diagram: allograft survival in kidney graft recipients stratified by absence or presence of BKPyV replication. Right-bottom diagram: allograft survival in patients with or without BKPyV-DNAemia. Right-upper diagram: DSA development in patients with or without BKPyV-DNAemia. The statistical difference between Kaplan–Meier survival curves was evaluated with the log-rank test; differences with *p* values < 0.05 were considered statistically significant. BKPyV: polyomavirus BK; IS: immunosuppression; CNI: calcineurin inhibitors; DSA: donor-specific antibodies.

**Figure 2 microorganisms-13-00048-f002:**
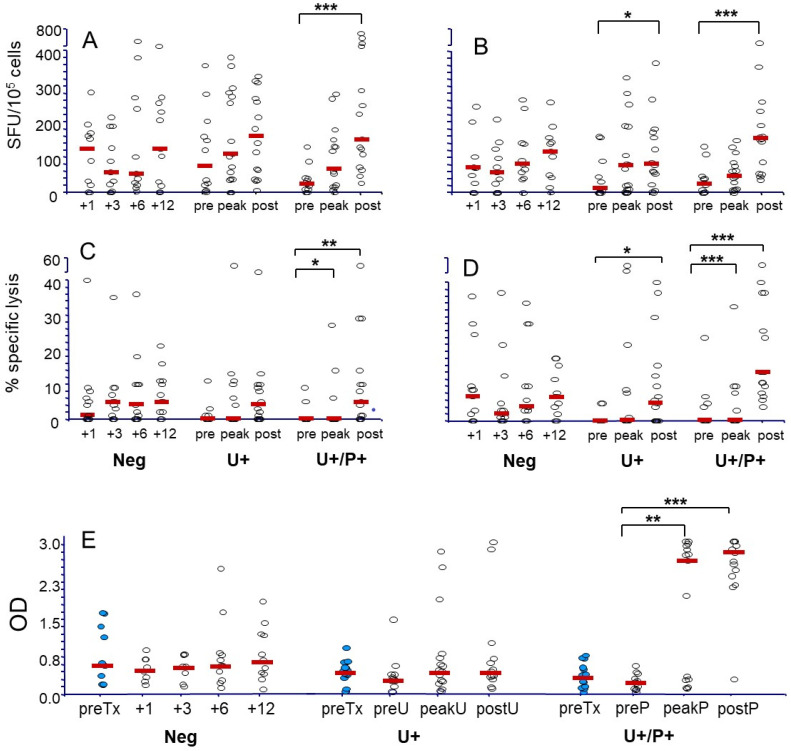
Kinetics of BKPyV-specific cellular and humoral immune responses in the KTx cohort. Data on the frequency of IFNγ-secreting lymphocytes (**A**,**B**) and virus-specific cytotoxicity (**C**,**D**), measured in patients’ cultured PBMCs, after a 10-day stimulation with VP1 (IFNγ-secreting lymphocytes: (**A**); cytotoxicity: (**C**)) and LT (IFNγ-secreting lymphocytes: (**B**); cytotoxicity: (**D**)) peptides, are reported. In (**E**), data on patients’ anti-BKV IgG are shown. Responses of KTx recipients who did not develop BKPyV-DNaemia and/or DNAuria (group 1, Neg), of DNAuric only (group 2, U+), and BKPyV-DNaemia-positive patients (U+P+, group 3) are reported. Blue dots represent pre-Tx data while white dots are for post-Tx measures; red rectangle represent median values. For groups 2 and 3, data obtained at the earliest time point before development of DNAuria (group 2) or DNAemia (group 3), at DNAuria or DNAemia increase, and at decrease and after viral clearance, are reported. IFNγ-secreting cells are represented as number of spots/10^5^ cells (median spots of triplicate experiments ± SD). Cytotoxicity is represented as % specific lysis at an effector to target ratio of 10:1 (mean of triplicate experiments ± SD). Differences among results obtained at the different time points were analyzed by the Kruskal–Wallis test (*: *p* < 0.05; **: *p* < 0.005; ***: *p* < 0.001).

**Figure 3 microorganisms-13-00048-f003:**
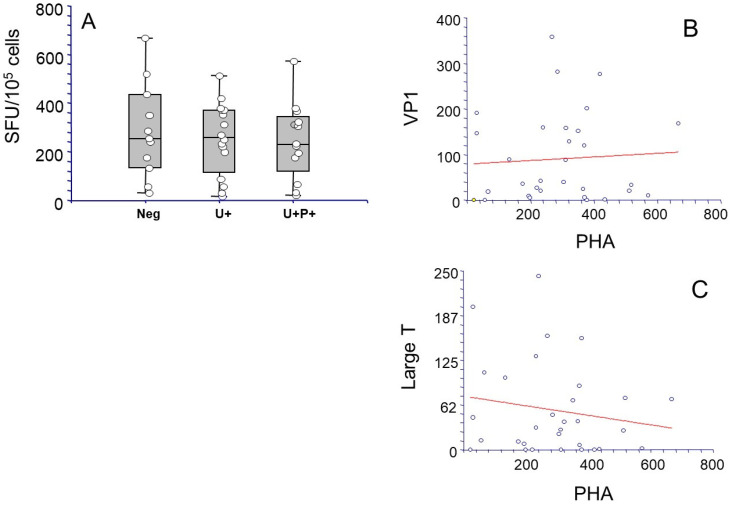
Immune response to polyclonal T cell activator PHA, and its relationship with BKPyV-specific immune response, in the KTx cohort. In (**A**), data on the frequency of IFNγ-secreting lymphocytes in response to PHA in KTx recipients who did not develop BKPyV-DNaemia and/or -DNAuria (group 1, Neg), of DNAuric only (group 2, U+), and of BKPyV-DNaemia-positive patients (U+P+, group 3) are reported. For group 1, assessment was at month + 1 post-KTx, while for groups 2 and 3, data were obtained at the earliest time point before development of viruria (group 2) or viremia (group 3). IFNγ-secreting cells are represented as number of spots/10^5^ cells (single patients as white dots, median spots of triplicate experiments ± SD as boxes). In (**B**,**C**), the correlation between response to PHA, as depicted in panel (**A**), and BKPyV VP1 and LT antigens at the same points of time, is reported.

**Figure 4 microorganisms-13-00048-f004:**
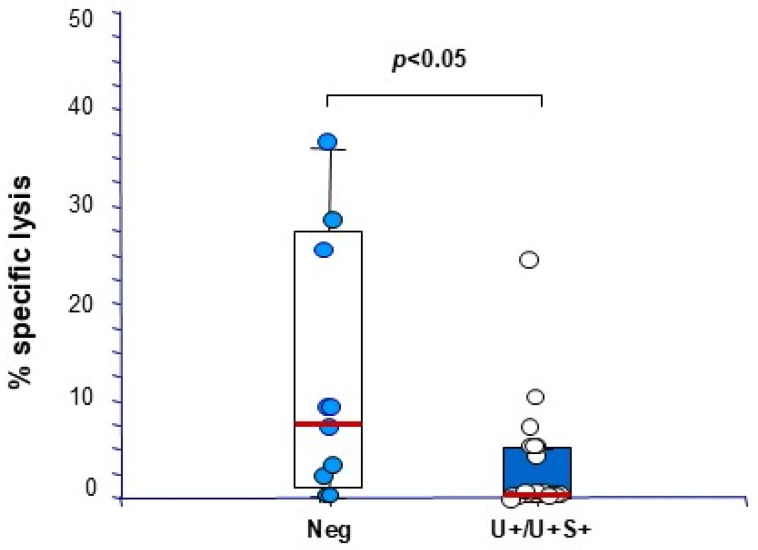
BKPyV LT-specific cytotoxicity in patients early after KTx. BKPyV LT-specific cytotoxicity, measured in samples obtained at 1 month after KTx in patients’ cultured PBMCs after 10-day stimulation with LT peptides, is reported. Responses of KTx recipients who did not develop BKPyV-DNAemia and/or DNAuria (group 1, Neg, blue dots), and of BKPyV-DNAuric only and DNAemic patients (U+/U+P+, groups 2 and 3, white dots) are reported. Cytotoxicity is represented as % specific lysis at an effector to target ratio of 10:1 (mean of triplicate experiments ± SD). Differences among results were analyzed by the Wilcoxon test. Red rectangle represent median values.

**Figure 5 microorganisms-13-00048-f005:**
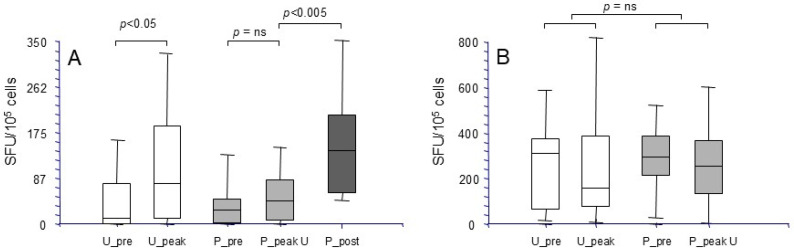
Kinetics of cellular immune responses in KTx recipients with BKPyV-DNAuria only versus BKPyV-DNaemia. Data on the frequency of IFNγ-secreting lymphocytes in response to LT peptides (**A**) or PHA (**B**) are reported. Responses of KTx recipients belonging to group 2 before (U_pre) and at peak (U_peak) DNAuria, and of group 3 DNAemic patients before replication (P_pre), at peak DNAuria (P_peak U) and at DNAemia clearance (P_post) are reported. IFNγ-secreting cells are reported as number of spots/10^5^ cells (median spots of triplicate experiments ± SD). Differences among results obtained at the different time points were analyzed by the Wilcoxon test; ns: not significant.

**Table 1 microorganisms-13-00048-t001:** Characteristics of the patients included in the study.

Patient-/Donor-/Transplant-Related Parameters	Numbers
Age (years), median (range)	16 (3–32)
Sex, M/F	28/17
No. of transplant, 1st/retransplant	33/12
Donor type, deceased/living	43/2
Donor age (years), median (range)	14 (2–57)
Baseline immunosuppression	
Any regimen + anti-CD25 mAb	40
Any regimen + ATG	3
CyA-based regimen	36
FK-based regimen	9
Any regimen + MMF	40
BKV serology, neg/pos	5/36
DGF	7
T cell-mediated rejection	9
CMV replication	21
EBV DNA positivity	23

CyA: cyclosporine-A; FK: tacrolimus; MMF: mycophenolate mofetil; DGF: delayed graft function.

**Table 2 microorganisms-13-00048-t002:** Risk of developing graft loss as a function of individual clinical parameters.

Variables	Patients (*n*)	HR *	95% CI	*p* Value
BKPyV positivity				
Yes	32	5.39	0.92–10.27	0.069
No	13			
BKPyV-DNAemia				
Yes	15	3.02	1.02–11.63	**0.047**
No	30			
Anti-HLA DSA				
Yes	20	6.7	1.72–19.46	**0.005**
No	25			
Antibody-mediated rejection				
Yes	16	24.28	5.05–58.77	**<0.0001**
No	29			
T cell-mediated rejection				
Yes	9	10.31	2.17–60.38	**0.004**
No	36			
Calcineurin inhibitor				
Tacrolimus	9	1.85	0.52–8.20	0.307
Cyclosporine-A	36			
PHA T cell immunity at early follow-up frequency > 360 SFU/10^5^ cells				
Yes	22	2.26	0.69–7.46	0.180
No	23			
BKPyV T cell immunity at early follow-up frequency > 177 SFU/10^5^ cells				
Yes	22	0.79	0.26–2.47	0.69
No	23			

* HR: hazard ratio.

## Data Availability

The clinical laboratory data of this study are available upon reasonable request from the corresponding author, according to privacy and ethical restrictions.

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
