# Peer review of "Control of BKPyV-DNAemia by a Tailored Viro-Immunologic Approach Does Not Lead to BKPyV-Nephropathy Progression and Development of Donor-Specific Antibodies in Pediatric Kidney Transplantation"

_microorganisms, 2024, doi:10.3390/microorganisms13010048_

Round 1

Reviewer 1 Report

Comments and Suggestions for Authors

Authors show an interesting study about BKV infection in kidney transplant recipients, presence of the virus in the urine and blood and related post transplant complications. Especially methods of BKV infection controlling, through T cell populations monitoring is of special importance.

Comments:

1) I suggest the change/shorten the title, jut highlighting novel method of BKV  infection monitoring/control,

2) please consider changing the abstract (L43-44): 30 patients developed viruria only (n=17), 

3) please explain all abbreviations, i.e. AST-IDCOP (L69), PCR (L132), ns (L187), Vp1, OD and others,

4) please change 'creatinine' into 'serum creatinine level' (L102 and later),

5) is acyclovir for 6 months and cotrimoxazole for 12 months standard prophylactic procedure? is it used in your centre or country? (L105-106),

6)  Figure 1:

- please omit 'solitary',

- in patients without BKV DNAemia (n=30) I suggest to change number of copies into < 1000,

-  in a group without BKV in the urine please specify that in the graph (not just no BKV),

- please explain all abbreviations in the Figure's caption,

7) please correct significance level p>0.0005 (L243),

8) please be more specific about 'anti-HLA class I or II antibodies' (L335 and later).

Comments on the Quality of English Language

Some words can be changed or corrected, i.e. 'persists' (L36), 'cause' (L58), 'group 1 patients' (L225).

Author Response

We wish to thank the reviewer for having substantially contributed to the revision, and, hopefully, improvement, of the manuscript.

1. I suggest the change/shorten the title, jut highlighting novel method of BKV infection monitoring/control.

We have changed the manuscript title, omitting the part on increased risk of graft loss, as suggested by the reviewer.

2. Please consider changing the abstract (L43-44): 30 patients developed viruria only (n=17).

As suggested by the reviewer, we changed the abstract (page 1, lines 44-45).

3.Please explain all abbreviations, i.e. AST-IDCOP (L69), PCR (L132), ns (L187), Vp1, OD and others.

As suggested, we checked for acronyms that were not expanded, and inserted proper definitions.

4. Please change 'creatinine' into 'serum creatinine level' (L102 and later).

We modified the text as requested by the reviewer.

5. Is acyclovir for 6 months and cotrimoxazole for 12 months standard prophylactic procedure? is it used in your centre or country? (L105-106).

The regimen described was a standard procedure for the centre at the time the transplants were performed.

6. Figure 1:

- please omit 'solitary',

- in patients without BKV DNAemia (n=30) I suggest to change number of copies into < 1000,

-  in a group without BKV in the urine please specify that in the graph (not just no BKV),

- please explain all abbreviations in the Figure's caption.

We have made the changes suggested by the reviewer.

7. Please correct significance level p>0.0005 (L243).

We corrected the significance level.

8. Please be more specific about 'anti-HLA class I or II antibodies' (L335 and later)

We expanded as suggested (page 10, lines 348-352).

Comments on the Quality of English Language

Some words can be changed or corrected, i.e. 'persists' (L36), 'cause' (L58), 'group 1 patients' (L225).

A native speaker has now corrected the English throughout the manuscript text.

Reviewer 2 Report

Comments and Suggestions for Authors

Thank you for reviewing this manuscript. The authors investigated the outcome of BKPyV nephropathy survival after pediatric kidney transplantation. They found that no significant correlation was observed between BKPyV-DNAemia and the development of DSA and antibody-mediated rejection. However, BKPyV-nephropathy plays a crucial role in kidney graft loss.

The study and its cohort are well structured, and the presentation and figures are clear. However, I found a few questions that should be answered.

First, the authors should address the patient’s follow-up time. The cohort seems to have been conducted quite 20 years ago.

The primary immunosuppression treatment after kidney transplantation has changed in recent years. Twenty years ago, the standard treatment was a CyA-based regimen, but nowadays, it is a tacrolimus-based regimen. Did the treatment change during the follow-up period?

However, perhaps the biggest problem is that the similarity is excessively high, so the authors should consider this to be corrected.

Author Response

We wish to thank the reviewer for having substantially contributed to the revision, and, hopefully, improvement, of the manuscript. 

1. The authors should address the patient’s follow-up time. The cohort seems to have been conducted quite 20 years ago.

The aim of the study was to assess the long-term impact of BKPyV infection (DNAemia, DNAuria and nephropathy) and its management. For this reason we reassessed the patients many years after transplant, looking also at DSA and ABMR development.

2. The primary immunosuppression treatment after kidney transplantation has changed in recent years. Twenty years ago, the standard treatment was a CyA-based regimen, but nowadays, it is a tacrolimus-based regimen. Did the treatment change during the follow-up period?

Stable patients without CyA-induced side effects were maintained in CyA, while other patients have been shifted to tacrolimus during follow-up. The shift, however, mostly occurred at a late post-transplant period, and did not impact BKPyV infection history, as BKPyV occurrence was generally an early event after KTx.

3. The similarity is excessively high, so the authors should consider this to be corrected.

We adjusted the title and the text in the Discussion section, and better explain that the observation of an impact on long-term outcome needs to be confirmed on a larger cohort, given the small size of the cohort (page 12, lines 427-429, and 432; page 13, line 482).